# Sulforaphane Inhibits Exhaustive Exercise-Induced Liver Injury and Transcriptome-Based Mechanism Analysis

**DOI:** 10.3390/nu15143220

**Published:** 2023-07-20

**Authors:** Jining Yang, Xinxin Guo, Tianyou Li, Yingquan Xie, Dawei Wang, Long Yi, Mantian Mi

**Affiliations:** Research Center for Nutrition and Food Safety, Chongqing Key Laboratory of Nutrition and Food Safety, Institute of Military Preventive Medicine, Third Military Medical University, Chongqing 400038, China

**Keywords:** exercise-induced liver damage, sulforaphane, glucose and lipid metabolism, Ppp1r3g, oxidative stress

## Abstract

Exhaustive exercise (EE) induces liver injury and has recently gained much attention. Sulforaphane (SFN) can protect the liver from inflammation and oxidative stress. However, the effects of SFN on EE-induced liver injury and its underlying mechanisms are still unclear. C57BL/6J mice swimming to exhaustion for seven days were used to simulate the liver injury caused by EE. Different doses of SFN (10, 30, 90 mg/kg body weight) were gavage-fed one week before and during the exercise. SFN intervention significantly reduced the EE-induced lactate dehydrogenase (LDH), creatine kinase (CK), alanine aminotransferase (ALT), and aspartate aminotransferase (AST) in the serum, as well as attenuating liver tissue morphological abnormality, oxidative stress injury, and inflammation. Liver transcriptomic analysis showed that the differentially expressed genes altered by SFN intervention in the exercise model were mainly enriched in glucose and lipid metabolism pathways. The most altered gene by SFN intervention screened by RNA-seq and validated by qRT-PCR is Ppp1r3g, a gene involved in regulating hepatic glycogenesis, which may play a vital role in the protective effects of SFN in EE-induced liver damage. SFN can protect the liver from EE-induced damage, and glucose and lipid metabolism may be involved in the mechanism of the protective effects.

## 1. Introduction

Various studies have high lightened the numerous health benefits of exercise of appropriate intensity, such as improving the body’s immunity, delaying aging, attenuating diabetes [1,2,3,4]. However, due to the requirements of some special occupations, such as marathon runners, extreme sports enthusiasts, and soldiers, they must carry out high-intensity or long-term exercise training. The potential health risks caused by vigorous exercise deserve special attention. Exercise-induced damage apparent in the sports system, such as bone fractures, muscle tearing, and tendon injury, are due to the intense symptoms and thus could get treated in time. Nevertheless, the symptoms of the injury of internal organs such as the liver are insidious; thus, exercise-induced liver damage is imperceptible and easily delayed. The liver is the largest secretory gland organ in the human body, participating in the metabolism of many nutrients and of the energy supply, thus playing an essential role in exercise performance [5]. Hepatocytes are rich in glycogen, which is an essential nutrient during intense exercise in consistently providing energy; at the same time, hepatocytes are also rich in mitochondria, which can generate amounts of ATP efficiently through the oxidative phosphorylation process for the cellular requirement. Studies have shown that vigorous exercise can cause acute liver injury characterized by alterations of liver enzymes in the serum [6], and liver morphological abnormality, oxidative stress, and inflammation were also detected in exhaustive exercise models of animals [7]. However, the mechanisms and potential targets underlying vigorous exercise-caused liver damage are still unclear; thus, there is a lack of prevention and treatment means.

Sulforaphane (SFN) is a natural isothiocyanate, the most common hydrolysis product of glucoraphanin distributed in Brassicaceae plants such as broccoli, kale, and radish. It is well known that SFN can induce the body to produce type II detoxification enzymes, such as heme oxygenase-1 (HO-1) and glutathione S-transferase (GSH), which makes it a powerful antioxidant in reducing excessive reactive oxygen species (ROS) in various tissues [8]. Recent studies have suggested that SFN could significantly improve exercise endurance and attenuate vigorous exercise-induced skeletal muscle damage and fatigue [9,10]. Moreover, the high bioavailability of SFN makes it a promising candidate for future clinical translation [11]. However, the previous studies usually inferred the mechanisms from the established ones of SFN in other disease models, such as the noted Nrf2-mediated pathway and the antioxidant effects [12,13,14]. Unbiased omics research methods have yet to be used for this issue, leaving the mechanisms underlying the potential protective effects of SFN on exercise-induced liver damage still need to be fully understood.

This study established the exercise-induced liver damage model through a swimming-to-exhaustive mouse for seven days. We examined the effects of SFN of different doses on the liver. RNA-seq has detected transcriptomic alterations, and bioinformatic analysis has been done to elucidate the underlying mechanisms and targets unbiasedly, thus providing a new intervention target for alleviating exercise-induced liver injury.

## 2. Materials and Methods

### 2.1. Chemicals and Reagents

SFN was purchased from MUST Biotechnology Co., Ltd. (Chengdu, China). IL-1b and TNF-a ELISA kits were purchased from Enzyme-linked Biotechnology Co., Ltd. (Shanghai, China). Mn-superoxide dismutase (SOD2), catalase (CAT), malondialdehyde (MDA), reduced glutathione (GSH), and oxidized glutathione disulfide (GSSG) assay kits were purchased from Beyotime Biotechnology Co., Ltd. (Shanghai, China).

### 2.2. Animals, Intervention Protocol, and Sample Collection

Ten-week-old healthy male C57BL/6J mice, weighing 24 ± 2 g, were purchased from LBT Laboratory Animal Co., Ltd. (Chongqing, China). The animals were housed under standard laboratory conditions (12 h on/off) and in a temperature and humidity-controlled environment (22–24 °C, 40–60%) in the Experimental Animal Center of the Army Medical University. Mice were allowed to move freely and given ad libitum access to a standard chow diet and water until the experiment started.

The intervention protocol and experimental groups are shown in Figure 1. Briefly, one week before the intervention, all mice were accustomed to the exercise by swimming for 10 min per day, and the mice that could not swim were eliminated. Then, mice were divided into six groups (n = 10): control, EXE, EXE + 10SFN, EXE + 30SFN, EXE + 90SFN, and 30SFN. The intervention period lasted for two weeks. During the first week (the gavage-fed week), mice were gavage-fed with SFN of different doses (10, 30, 90 mg/kg BW) or the solvent as the control. In the second week (the exercise week), mice were subjected to swimming to exhaustion for seven days or just moving freely as a sedentary control. Mice swam in a round tank (100 cm in diameter, 50 cm deep) with 33–36 °C water, as suggested [15]. The exhaustion time point was determined when the mice failed to reach the water’s surface for 5 s and blunted righting reflex according to the Thomas and Marshall criteria [16]. After swimming, animals were dried, kept warm, and returned to their cages.

The sedentary groups were sacrificed 2 h after oral ingestion, and the exercise groups were sacrificed immediately after exhaustion. Blood samples were collected rapidly from the orbital venous plexus and centrifuged at 1500× *g* for 15 min to collect the serum. The liver tissues were collected. The left lobes of the liver tissue were fixed in 4% paraformaldehyde for histopathological examination; the other parts of the liver tissue were immediately stored in liquid nitrogen and then transferred to a −80 °C refrigerator.

### 2.3. Serum Biochemical Parameters

Lactate dehydrogenase (LDH), creatine kinase (CK), alanine aminotransferase (ALT), and aspartate aminotransferase (AST) in the serum were measured by an automatic biochemical analyzer (2110, Hitachi, Japan). The ELISA kits were used to detected TNF-α and IL-1β in the serum according to the manufacturer’s instructions.

### 2.4. H&E Staining

Liver tissue fixed in 4% paraformaldehyde was dehydrated, embedded in paraffin wax, and then sectioned into 5 µm-thick slices. The sections were deparaffined and rehydrated via a xylene and ethanol series before performing H&E staining according to the standard histology procedure previously described [17].

### 2.5. Detection of Oxidative Stress

The levels of MDA, SOD2, CAT, and GSH/GSSH in the liver homogenates were detected according to the manufacturer’s instructions for the assay kits. Briefly, the supernatants of the liver homogenates lysed by RIPA were collected and reacted with respective assay reagents. The absorbance at the specific wavelength was measured using a microplate reader (Molecular Devices, San Jose, CA, USA). At the same time, the protein level of each sample was measured via BCA assay. The level of MDA was displayed as µmol/mg protein, and the activities of SOD2 and CAT were displayed as Unit/mg protein.

### 2.6. RNA Sequencing and Bioinformatic Analysis

Liver tissues of the four groups (control, EXE, EXE + 30SFN, 30SFN) were collected, and 4 samples of each group were randomly selected for the RNA sequencing analysis. RNA was isolated by TRIzol reagent (Invitrogen, Waltham, MA, USA), as described in the instructions, and the RNA quality was determined by examining A260/A280 with a Nanodrop^TM^ 3000 (ThermoFisher Scientific Inc., Waltham, MA, USA). RNA integrity was confirmed by 1.5% agarose gel electrophoresis, and the qualified RNAs were finally quantified by Qubit 3.0 with a Qubit^TM^ RNA Broad Range Assay kit (Life Technologies, Carlsbad, CA, USA). A total of 2 µg of total RNAs were used for stranded RNA sequencing library preparation using the KC-Digital^TM^ Stranded mRNA Library Prep Kit for Illumina^®^ (Seqhealth Inc., Wuhan, China). The kit eliminates duplication bias in PCR and sequencing steps by using a unique molecular identifier (UMI) of 8 random bases to label the pre-amplified cDNA molecules. The library products corresponding to 200–500 bps were enriched, quantified, and finally sequenced on a Novaseq 6000 sequencer with a PE150 model.

Sequencing reads that were of low quality or contained only adapters were pre-filtered, and then the rest of the sequences were mapped to the mm10 mouse genome using a STAR RNA-Seq aligner [18]. Reads mapped to the reference were counted by featureCounts [19], and then reads per kilobase per million mapped reads (RPKMs) were calculated. Differentially expressed genes (DEGs) between groups were detected with edgeR [20], with the criteria of adjusted *p*-value (FDR) < 0.05. The heatmap of the DEGs was generated using the heatmap R package [21]. Gene Ontology (GO) and the Kyoto Encyclopedia of Genes and Genomes (KEGG) enrichment analysis of DEGs were analyzed by KOBAS 3.0 [22] and displayed using the ggplot2 R package [23]. Gene set enrichment analysis (GSEA) was used to investigate potential mechanisms in the Molecular Signatures Database (MSigDB) [24,25]. The RNA-seq data entitled “Transcriptome studies of liver tissue in exhaustive exercise-induced mouse model with or without SFN intervention” were deposited to the Sequence Read Archive (SRA) with BioProject number PRJNA985800.

### 2.7. RNA Isolation and qRT-PCR

Liver tissues were harvested in RNAiso Plus reagent (Takara Bio, Shiga, Japan), and the total RNA was extracted according to the manufacturer’s instructions. Then, the RNA concentration and purity were measured by a NanoDrop^TM^ 2000 spectrophotometer (ThermoFisher Scientific Inc., MA, USA). The first-strand cDNA was synthesized by a PrimeScript RT Master Mix kit (Takara Bio, Shiga, Japan). qRT-PCR was carried out with the qTower 2.2 real-time PCR system (Anakytik Jena, Thuringia, Germany) using SYBR Premix Ex Taq II (Takara Bio, Shiga, Japan). The primers for the targeted genes were synthesized by Sangon Biotech (Shanghai, China). The primer sequences used for gene expression analysis are listed in Appendix A. The PCR protocol included 3 steps: denaturation at 95 °C for 30 s, followed by 40 cycles of 95 °C for 5 s and 60 °C for 30 s. A melting curve protocol was run at the end of the amplification. Relative fold changes in gene expression were analyzed by the 2^−ΔΔCt^ method and normalized to the internal control gene Gapdh [26].

### 2.8. Statistical Analysis

The data analysis from RNA sequencing is described in the corresponding section. The other data of the article were expressed as mean ± SD and analyzed with SPSS 22.0 software (Chicago, IL, USA). The normality of study data was checked by the Shapiro–Wilk test. The significance of the normally distributed data was analyzed by one-way or two-way analysis of variance (ANOVA), followed by Tukey’s test. *p*-values less than 0.05 were considered statistically significant.

## 3. Results

### 3.1. Sulforaphane Intervention Combined with EE Slowed down the Age-Dependent Body Weight Gain in Mice

The body weight of the mice was measured at three time points: before the gavage-fed (time point 1), before the EE (time point 2), and after the EE (time point 3), and the results are shown in Table 1. There was no significant difference in the body weight of the mice among the groups in any of the three time points. However, during the whole intervention time, there was an overall elevated trend in the body weight of the mice (*p* = 0.002 for time point 1 vs. time point 2, *p* < 0.0001 for time point 1 vs. time point 3, *p* < 0.0001 for time point 2 vs. time point 3). Control and 30SFN groups showed significant age-dependent body weight gain: the body weight of time point 3 was higher than time point 2 and time point 1. However, there was no significance in body weight between time point 3 and time point 2 in EXE and EXE + 10SFN groups; the overall elevation of body weight was not detected in EXE + 30SFN and EXE + 90SFN groups. The results suggest that mid and high doses of sulforaphane intervention combined with EE could slow the age-dependent growth of mice’s body weight.

### 3.2. Sulforaphane Improved Exercise Performance and Fatigue Biochemical Parameters in Mice Subjected to EE

Swimming time is commonly measured as an index of exercise performance because mouse swimming constitutes treading water in a limited area, and the swimming distance is not measured or useful [15]. A longer swimming time indicates better exercise endurance and less susceptibility to fatigue. During the seven days of EE, every mouse’s swimming time to exhaustion was measured, and the results are shown in Figure 2a. In the EXE group, the time to exhaustion showed a downward trend in the exercise week for seven consecutive days, while there was no significant downward trend in the groups intervened by SFN. Through analysis by two-way ANOVA, the times to exhaustion of the three doses of the SFN intervention group were all significantly longer than that of the EXE group (*p* < 0.05 for the three comparisons); time to exhaustion was slightly longer in the EXE + 30SFN and EXE + 90SFN than in the EXE + 10SFN group (*p* < 0.05 for the two comparisons), but the time was not statistically different between the EXE + 30SFN and EXE + 90SFN groups *(p* = 0.2066). LDH and CK in the serum can be used to evaluate the degree of fatigue [27,28]. The concentration of LDH and CK were significantly increased in the EXE group compared with the control group but decreased with the all the three doses of SFN intervention (Figure 2b,c). No significance was detected in the LDH and CK levels between the EXE + 30SFN and EXE + 90SFN groups (*p* > 0.9999 in LDH, and *p* = 0.941 in CK). The above results suggest that SFN intervention can significantly enhance exercise performance and reduce fatigue in mice.

### 3.3. Sulforaphane Attenuated EE-Induced Inflammation and Liver Enzyme Elevation

Elevated serum ALT and AST are typical signs of liver diseases, suggesting hepatocyte damage or necrosis [29]. In the serum, inflammatory biomarkers TNF-α and IL-1β are commonly used to reflect the body’s inflammation. As shown in Figure 3, ALT, AST, TNF-α, and IL-1β were elevated significantly in the EXE group compared with the control group. SFN intervention at all doses significantly reduced serum AST and TNF-α. IL-1β decreased in mid- and high-dose SFN intervention, and ALT only decreased in SFN treated with mid-dose. These results suggest that SFN intervention can attenuate EE-caused liver cell damage and necrosis, triggering inflammation.

### 3.4. Sulforaphane Improved EE-Induced Liver Pathological Morphology Abnormality

Pathological morphology of the liver was observed through H&E staining. The representative images of H&E staining are shown in Figure 4. In the groups that were not subjected to EE, i.e., the control group and the 30SFN group, the structure of the liver was intact without noticeable swelling, rupture, and inflammatory cell infiltration, while the liver of the mice subjected to EE showed noticeable cell swelling, unclear cell membrane boundary, tissue damage, and unclear liver cord. Although cell swelling and inflammatory cell infiltration were presented in the liver of the mice intervened by SFN, the degree of damage was lighter than that of the EXE group, and collagen hyperplasia could be observed in the damaged area of liver cells, suggesting that the liver was recovering after the injury.

### 3.5. Sulforaphane Attenuated EE-Induced Liver Oxidative Stress

During vigorous exercise, the energy requirement increases rapidly; thus, the oxidative phosphorylation that produces ATP occurs frequently, accompanied by its by-product, reactive oxygen species (ROS). Excessive ROS can cause oxidative stress to target tissues. The lipid peroxidation substrate glutaraldehyde (MDA) is an essential indicator of the degree of oxidative stress damage, while SOD2, CAT, and the ratio of GSH to GSSH reflect the ability to resist oxidative stress. Assay kits were used to detect the above indicators in the liver tissue, and the results are shown in Figure 5. The level of MDA in the liver increased significantly in the EXE group, suggesting that severe oxidative stress damage occurred in the liver. At the same time, the intervention of SFN in the three doses inhibited the increase in MDA (Figure 5a). Compared with the control group, the activity of SOD2, CAT, and the ratio of GSH/GSSG decreased significantly in the EXE group; compared with the EXE group, the intervention of mid- and high-dose of SFN could significantly increase the activity of SOD2 and the ratio of GSH/GSSG, and the intervention of high concentrations of SFN could increase the activity of CAT (Figure 5b–d). The above results suggest that SFN intervention can promote the antioxidant capacity in the liver, thus attenuating EE-induced liver oxidative stress, and the high dose of SFN showed generally better effects than the moderate and low doses.

### 3.6. Liver Transcriptome Alterations in the SFN Intervened EE Mice Model

To investigate the underlying mechanism of SFN protecting against EE-induced liver injury, we detected and analyzed the transcriptomes of the liver by RNA sequencing. As the experiment results showed, the mid-dose of SFN induced most of the protection effects; thus, we detected the transcriptomes of the mid-dose of SFN intervention and the control groups, i.e., control, EXE, EXE + 30SFN (abbreviated as EXE + SFN in the transcriptome analysis), and 30SFN.

The principal component analysis (PCA) of the sequencing results is shown in Figure 6a; the groups that were not subjected to EE (control and 30SFN group) are found in the lower part of the coordinate chart, and the samples of the two groups are interleaved, which cannot be clearly distinguished into two groups, indicating that the transcriptome gene expression patterns of the two groups are relatively similar. The EXE group is on the upper left of the coordinate plot, and the EXE + SFN group is located on the upper right of the coordinate chart, which is clearly distinguished from the other three groups, indicating that the EE and SFN interventions significantly changed the transcriptome of the mouse liver.

Further, DEGs between groups were screened out by FDR < 0.05 and log2|FC| >1 criteria. Compared with the control group, the expression levels of 467 genes were elevated, and 252 genes were decreased in the EXE group. Compared with the EXE group, the expression levels of 725 genes were elevated, and 924 genes were decreased in the EXE + SFN group. However, only 15 genes were elevated, and 7 genes were deceased when 30SFN was compared to the control group. All the expressions of the DEGs are displayed as a heatmap in Figure 6b–d. These results suggest that EE and SFN intervention significantly affected the transcriptomes of liver and SFN intervention.

### 3.7. Pathway Enrichment Analysis of the DEGs of Different Comparison Groups

To explore the potential biological changes underlying the transcriptome alterations, we performed GO and KEGG enrichment analyses of the DEGs, as shown in Figure 6b,c. Taking FDR < 0.05 as the enrichment standard, the enrichment results of the upregulated and downregulated DEGs in EXE compared to the control are shown in Figure 7a. The enriched GO items in the EXE group were mainly concentrated in fat biosynthesis. At the same time, the genes downregulated in the EXE group were mainly enriched in fatty acid oxidation, defense response, nitric-oxide synthase regulator activity, etc., suggesting that EE might have an important impact on fatty acid metabolism and lead the body to a deleterious state, such as reduced defense response and reduced antioxidant ability. Comparing EXE with the EXE + SFN group, the enriched GO terms of the DEGs are shown in Figure 7b. The upregulated genes were enriched in items such as catabolism and autophagy. In contrast, the downregulated genes were enriched in the items such as inflammatory response activation and oxidase activity. The results suggest that SFN intervention can reduce the EE-induced liver inflammatory response and promote metabolism at the transcription level.

Since the changing direction of genes in the pathway is not necessarily the same, both the up- and downregulated DEGs were used in the enrichment analysis of the KEGG pathway. As shown in Figure 7c,d, the DEGs of the EXE and control groups were mainly enriched in metabolic pathways, especially lipid and carbohydrate metabolism, and the DEGs between the EXE and EXE + SFN groups were enriched in the immune system and signal transduction pathways in addition to cholesterol and glycogen metabolism. The above results suggest that EE induced a significant change in metabolism pathways of the liver transcriptome, and SFN intervention impacts metabolism and immunity.

### 3.8. Gene Set Enrichment Analysis (GSEA) of the Liver Transcriptomes

Based on the above GO and KEGG results, we performed GSEA on the two comparison groups in the Hallmark dataset of the Molecular Signatures Database (MSigDB) [30]. The GSEA algorithm calculates an enrichment score reflecting the degree of overrepresentation at the top or bottom of the ranked list of the genes included in a gene set in a ranked list of all genes in the RNA-seq dataset. A positive enrichment score (ES) indicates gene set enrichment at the top of the ranked list; a negative ES indicates gene set enrichment at the bottom of the ranked list [24]. Three representative items related to inflammation and metabolism, namely, TNF-α signaling via NFκb, oxidative phosphorylation, and fatty acid metabolism, were selected and enriched between the two comparison groups, and the results are shown in Figure 8. TNF-α signaling via NFκb, which represents inflammation, was positively correlated with the EXE group when compared with the control and EXE groups (Figure 8a) and remained positively correlated with the EXE group when compared with the EXE + SFN group (Figure 8d). The oxidative phosphorylation was positively correlated with the EXE group when compared with the control group (Figure 8b) and positively correlated with the EXE + SFN group when compared with the EXE group (Figure 8e). The fatty acid metabolism was positively correlated with the EXE group when compared with the control group (Figure 8c) and positively correlated with the EXE + SFN group when compared with the EXE group (Figure 8f). These results suggest that SFN intervention can attenuate the EE-induced inflammation in the liver and further promote oxidative phosphorylation and fatty acid metabolism.

### 3.9. Ppp1r3g Is a Candidate Gene for the Protective Effect of Sulforaphane on EE-Induced Liver Injury

In order to find the potential key genes for SFN in the protective effects on EE-induced liver injury, the mutual DEGs of EXE vs. control and EXE + SFN vs. EXE were identified using Venn analysis, shown in Figure 9a. We focused on the genes that were elevated in EXE but decreased in the intervention of SFN and the genes that decreased in EXE but increased in the intervention of SFN. These genes were then ranked based on the foldchange, and the top 10 are displayed in Table 2, the most significant of which is Ppp1r3g, one of the catalytic subunits of the protein phosphatase 1 (PP1) holoenzyme [31]. qRT-PCR was then performed to validate the expression of the candidate gene, and the alteration trend was consistent with the RNA-seq result (Figure 9b,c). The other nine genes listed in Table 2 were also validated by qRT-PCR, the results of which are displayed in Appendix A. The qRT-PCR results of six genes (Tas1r2, Fbp2, Nr4a2, Gm11525, Hdc, and Gtse1) are consistent with the RNA-seq results. The expression of the other three genes (Crtam, Tex35, and Wnt6) showed no or less significant differences among groups, though they are in the same changing trend, suggesting that results of RNA-seq and qRT-PCR are in good consistency. Also, the results suggest that Ppp1r3g is the potential key gene of the protective effect of SFN on EE-induced liver damage.

## 4. Discussion

Exercise significantly impacts human health, but whether it benefits or is harmful to the body largely depends on the intensity. It has been well proven that moderate exercise exerts significant good effects on different organs, such as attenuating metabolic disorders, but vigorous exercise could be detrimental even to the healthy body [32,33]. The liver is a metabolic organ that plays a crucial role in scavenging ROS, synthesizing proteins, metabolizing fatty acid, and storing glycogen, which is closely related to exercise, making it a target for the damage caused by excessive exercise. Studies have reported that acute exercise could cause abnormal liver enzyme profile [6], and exhaustion exercise could even induce severe liver tissue pathological abnormality and liver inflammation [7]. In this study, mice were subjected to exhaustion exercise for seven consecutive days, and similar liver injuries were observed: AST and ALT were significantly increased, and liver pathology was abnormal, indicating that the EE-induced liver injury model had been successfully established. On the other hand, the liver is rich in glycogen and mitochondria, thus generating energy through oxidative phosphorylation rapidly during exercise [34]. Therefore, liver damage could further weaken exercise performance. Here, we found that the fatigue markers, LDH and CK, were significantly increased in mice with EE. The time to exhaustion declined during the consecutive seven EE days, along with which the body weight of the mice in the EXE group did not increase as in the control group, suggesting that the consecutive EE increased fatigue and reduced exercise endurance in mice.

ROS is a by-product of oxidative phosphorylation during EE, and the rapid ROS accumulation can cause oxidative stress damage to target tissues [35]. Usually, ROS in the body is in a low or moderate concentration of equilibrium, which can be a signaling molecule participating in the transduction of signaling pathways. Physiologically, the redox systems in our body maintain redox homeostasis, limiting ROS to a proper level [36]. However, the ROS level could rise fast when exposed to detrimental stimuli, for example, exhaustive exercise; thus, the capability of the anti-oxidative system that removes ROS has a significant impact on the severity of the detrimental stimuli caused damage. SOD2, CAT, and GSH/GSG belong to the ROS scavenging system. MDA is the product of lipid peroxidation and is a sensitive indicator of oxygen radical metabolism in the body. The results of this study showed that EE caused a significant increase in the level of oxidative stress in the liver of mice and a significant decrease in antioxidant capacity. The pre-intervention of SFN significantly increased the activity of antioxidant enzymes in the liver and reduced the oxidative stress damage of the liver.

Sulforaphane is an effective inducer of Nrf2, a key transcription factor in triggering strong cellular defense response and anti-oxidative stress. Due to the variety of sources and relative safety to the human body, food-derived phytochemicals are the potential candidate for preventing and treating many diseases, including EE-induced injuries. Previous studies have reported that some polyphenols could improve post-exercise recovery [37]. Quercetin, for example, a kind of flavone, plays a role in protecting against excessive exercise-induced damage through its anti-inflammatory effects [7]. However, the low bioavailability of these phytochemicals limited the translation to clinical use. This study used SFN, a kind of isothiocyanate with high bioavailability, as the intervention compound to investigate the potential protective effects and mechanisms in EE-induced liver damage, which is significant in the future translation. Here, we found that pre-intervention of SFN in EE mice improved the capability of anti-oxidative stress and reduced inflammation in the liver, thereby attenuating liver damage and improving exercise endurance. Interestingly, we also found a slowdown effect on body weight gain in the combined intervention of mid or high doses of SFN and EE, which is more significant than the intervention of SFN or EE alone. According to the toxicology research, the LD50 value of SFN in mice was estimated to be 213 mg/kg body weight [38], which is much higher than the doses used in the present study. It seems that SFN and exercise have a synergistic effect on body weight gain. A recent clinical study has reported that the training combined with broccoli supplementation, a vegetable rich in SFN, could decrease the BMI in male adults with diabetes, while the training alone or broccoli supplementation alone exerted no such effects compared with the control group [39]. It has been well reported that SFN could reduce obesity through many mechanisms, such as reversing leptin resistance [40], browning the white fat [41], and promoting lipolysis [42]. However, few studies report the effects of the combination of SFN and exercise on healthy people’s body weight. The results of our animal study provided a concern that additional energy reinforcement might be required to maintain weight when SFN is used to protect against EE-induced liver injury in future translational studies.

Further, the study tried to screen the most likely underlying mechanism of the protective effects of SFN in EE-induced liver damage by an unbiased strategy. Here, digital (UID) RNA-seq [43] was used to identify the transcriptome, and bioinformatic analysis was used to predict candidate genes or pathways. It can be inferred from the principal component analysis that EE significantly changed the overall transcriptome of the liver, and SFN intervention regulates the transcriptome significantly in the case of EE rather than the control. The PCA results were in accordance with the numbers of DEGs to some extent: only 15 DEGs were detected between the control and 30SFN, which is far from the thousands of differential genes between other comparison groups. GO and KEGG enrichment analysis showed that genes changed under the stimulation of EE and SFN intervention, respectively, changed enrichment in different pathways. Compared with GO and KEGG enrichment analysis, GSEA enrichment analysis does not require pre-screening of DEGs but finds the gene collection with synergistic variance from the expression matrix of the genes, so the genes with less difference can also be considered. Based on the preliminary results of GO and KEGG enrichment, we selected three predefined gene sets for GSEA enrichment and found that the gene expression alterations caused by exercise and SFN intervention, respectively, showed opposite changes in inflammation-related gene sets but were consistent in oxidative phosphorylation and fatty acid metabolism. The results suggest that high-intensity exercise promotes the inflammatory response, and SFN intervention can ultimately suppress the inflammatory response. On the other hand, SFN further increases some adaptive reactions caused by exercise [44], such as oxidative phosphorylation and fatty acid metabolism, making the body more efficient in providing energy during exercise.

In this study, 240 DEGs were screened out through the specific method that could represent the potential target of SFN in protecting EE-induced liver damage. RT-qPCR was further used to validate the top 10 DEGs among the 240 DEGs. Among them, the Ppp1r3g is the DEG with the most considerable fold change when compared with the control group to the EXE group, from which could be inferred that this gene played an important role in the protective effect of EE-induced liver damage. Our result showed that the expression level of Ppp1r3g increased with the stimulation of EXE and decreased with the SFN intervention. PPP1R3G is one of the regulatory subunits of protein phosphatase 1 (PP1) holoenzyme, which is cell specific and thus specific to protein substrates within different tissues and cells [45,46]. The PPP1R3 protein family contains seven members (PPP1R3A-G) that can target PP1 to glycogen metabolic pathways by binding glycogen synthase, glycogen phosphorylase, or glycogen phosphorylation kinase, and finally phosphorylate the substrate through the catalytic subunit of PP1 [47,48]. Previous studies found that mice with PPP1R3G knock-out showed a phenotype of reduced glycogen storage capacity in the liver [49]. A recent study suggested that the underlying mechanism might be that the protein directly phosphorylates its substrate, serine/threonine kinase (AKT), thereby regulating glycogen synthesis and thus maintaining blood glucose homeostasis [50]. In our study, the expression of the Ppp1r3g gene in the liver was increased by exhaustive exercise and reversed with the SFN intervention, suggesting that the Ppp1r3g-mediated glycogen homeostasis may be a potential mechanism for SFN to protect against high-intensity exercise injury. Although our transcriptome-based mechanistic analysis study found an interesting potential target gene, further experimental investigation should be done, and there is still a long way to go to translate the preliminary findings into clinical application.

In summary, sulforaphane can improve the antioxidant capacity of the liver, reduce the oxidative stress damage of the liver caused by high-intensity exercise, improve the liver enzyme profile and systemic inflammatory response, promote fatigue recovery, and improve exercise tolerance; transcriptome studies suggest that the mechanism may be related to liver glycogen metabolism homeostasis and fatty acid metabolism homeostasis, and Ppp1r3g may play a key role in the effects.

## 5. Conclusions

Sulforaphane can attenuate exhaustive exercise-induced liver injury and improve exercise performance. The underlying mechanisms revealed by RNA-seq suggest that Ppp1r3g-mediated metabolic regulation may play an important role.

## Figures and Tables

**Figure 1 nutrients-15-03220-f001:**
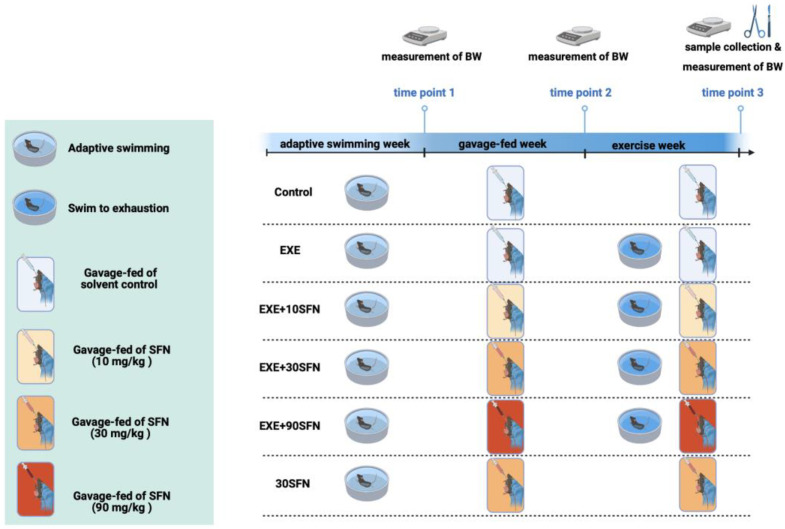
Schematic diagram of experimental design.

**Figure 2 nutrients-15-03220-f002:**
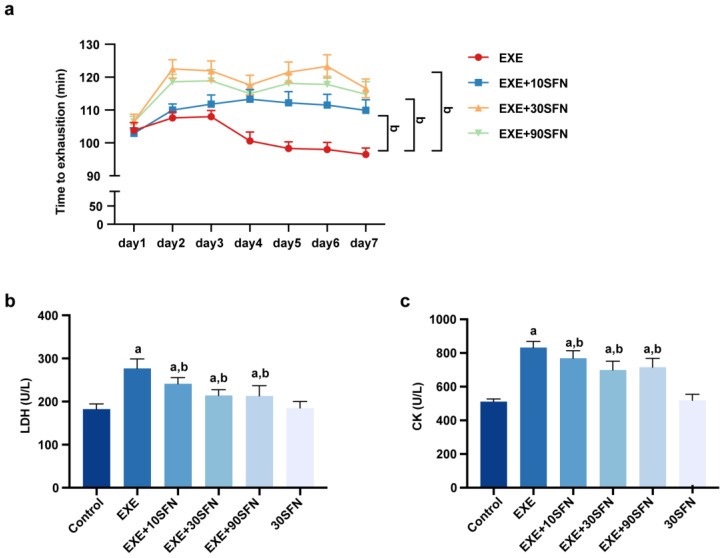
Sulforaphane improves exercise performance and biochemical fatigue parameters in mice subjected to EE. (**a**) Time to exhaustion during the exercise week. (**b**,**c**) LDH and CK levels in the serum of mice after the exercise week (^a^: *p* < 0.05 vs. control group, ^b^: *p* < 0.05 vs. EXE group).

**Figure 3 nutrients-15-03220-f003:**
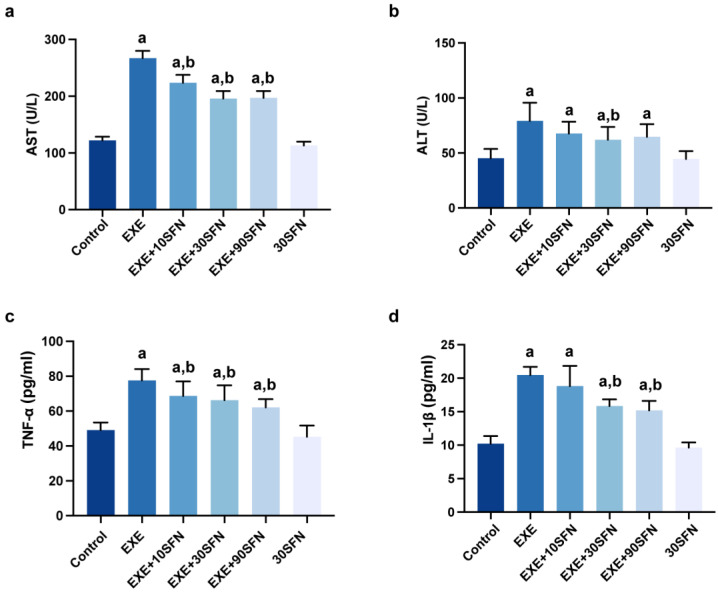
Sulforaphane attenuated EE-induced inflammation and liver enzyme elevation. (**a**,**b**) AST and ALT levels in the serum of mice after the exercise week. (**c**,**d**) The inflammation factors, TNF-α levels and IL-1β levels, in the serum of mice after the exercise week. (^a^: *p* < 0.05 vs. control group, ^b^: *p* < 0.05 vs. EXE group.)

**Figure 4 nutrients-15-03220-f004:**
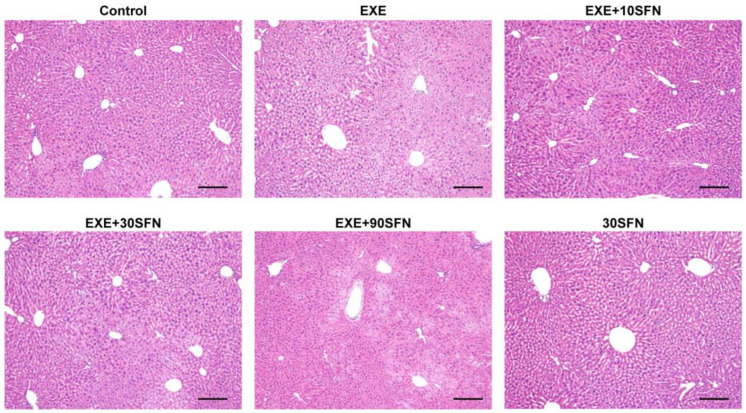
The presentative images of the H&E staining of the liver after the exercise week. Magnification: 100×.

**Figure 5 nutrients-15-03220-f005:**
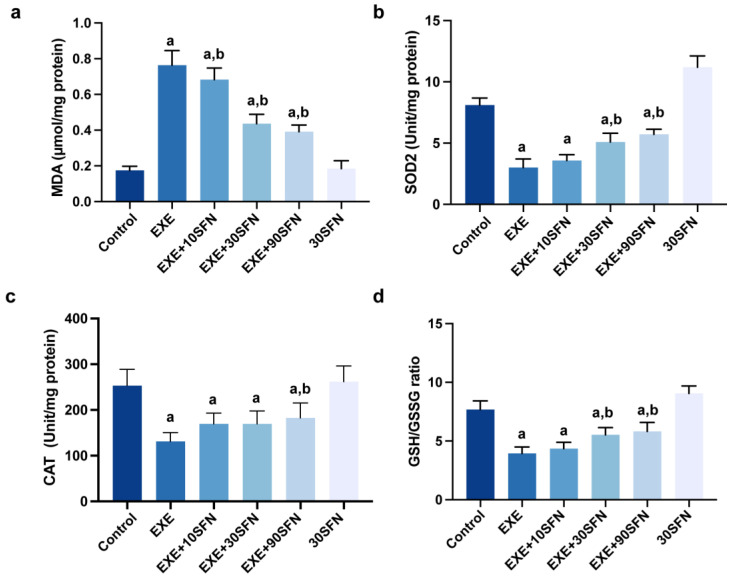
Sulforaphane attenuated EE-induced liver oxidative stress. (**a**–**d**) The MDA, SOD2, and CAT levels and GSH/GSSG ratio of the liver tissue after the exercise week. (^a^: *p* < 0.05 vs. control group, ^b^: *p* < 0.05 vs. EXE group.)

**Figure 6 nutrients-15-03220-f006:**
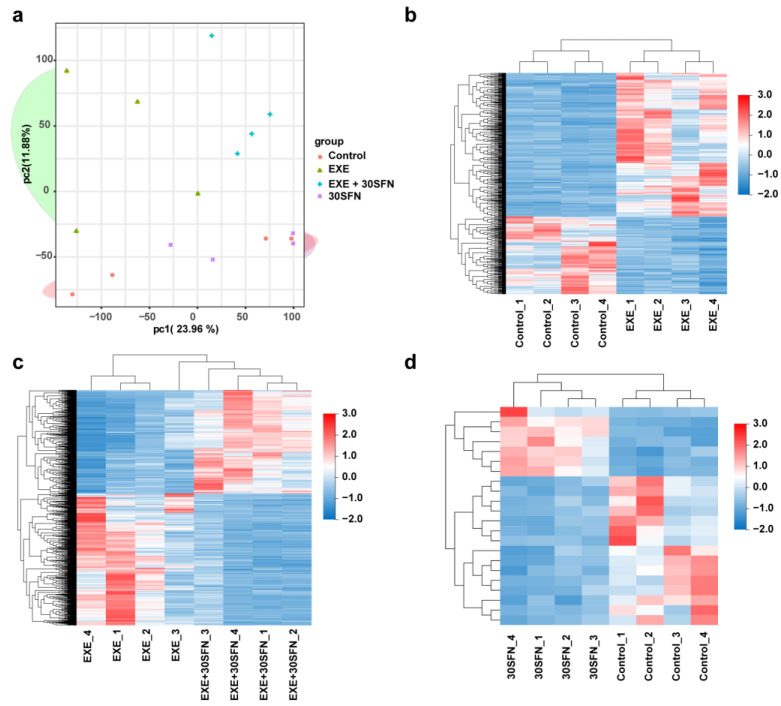
Transcriptome alterations of liver tissues of each group. (**a**) PCA of the gene expression of the control, EXE, EXE + SFN, and 30SFN samples. (**b**) The heatmap of DEGs between the control and EXE groups. (**c**) The heatmap of DEGs between EXE and EXE + SFN. (**d**) The heatmap of DEGs between 30SFN and the control.

**Figure 7 nutrients-15-03220-f007:**
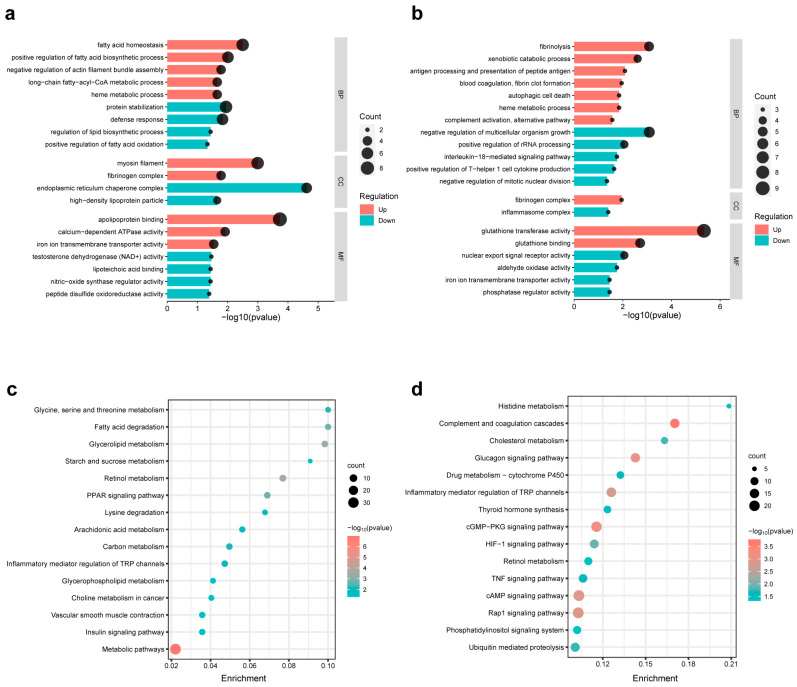
GO and KEGG pathway analysis of DEGs. (**a**,**b**) Significantly enriched biological process (BP), cell component (CC), and molecular function (MF) of GO terms of the DEGs between control and EXE (**a**) and DEGs between EXE and EXE + SFN (**b**): up- (in the red shade in the figure) or down- (in the blue shade in the figure) regulated by EXE (**a**) or EXE + SFN (**b**). In the vertical axis are the terms, and in the horizontal axis are the transformed FDR (−log_10_Pvalue). (**c**,**d**) Bubble chart showing enriched KEGG pathways of the DEGs between the control and EXE (**c**) and DEGs between EXE and EXE + SFN (**d**). The horizontal axis represents the ratio of the number of input genes to the background gene, and the vertical axis represents the pathways. The size of the dots represents the number of enriched genes, and the shades of color represent the transformed FDR (−log_10_Pvalue).

**Figure 8 nutrients-15-03220-f008:**
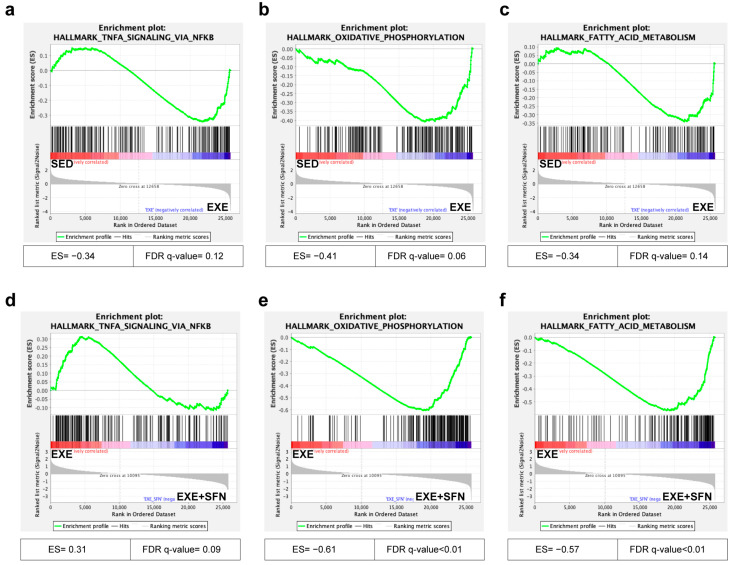
The GSEA results of the control and EXE groups and EXE and EXE + SFN groups. (**a**–**c**) GSEA was performed in the control and EXE groups. The TNF-α signaling via NFκb (**a**), oxidative phosphorylation (**b**), and fatty acid metabolism (**c**) was enriched in the EXE group. (**d**–**f**) GSEA was performed in the EXE and EXE + SFN groups. The TNF-α signaling via NFκb (**d**) was enriched in the EXE group, while oxidative phosphorylation (**e**) and fatty acid metabolism (**f**) were enriched in the EXE + SFN group.

**Figure 9 nutrients-15-03220-f009:**
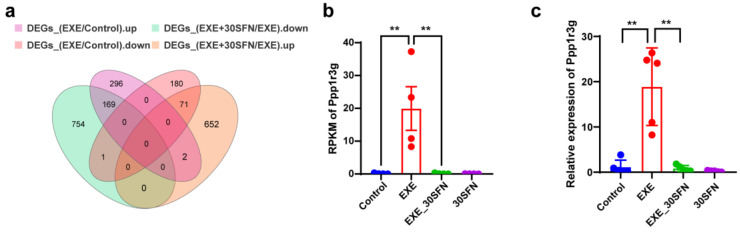
Candidate gene screening and validation. (**a**) The Venn diagram illustrating the common and different DEGs of different comparisons of groups. (**b**) The RPKM of Ppp1r3g was detected by RNA sequencing. (**c**) The relative expression of Ppp1r3g was determined by qRT-PCR. (**: *p* < 0.01.)

**Table 1 nutrients-15-03220-t001:** Body weight at different time points.

Group	Before Gavage Fed (g)	Before the EE (g)	After the EE (g)
Control	25.35 ± 0.63	25.70 ± 1.18	27.46 ± 1.87 *^,#^
EXE	25.61 ± 0.80	26.55 ± 0.71	27.41 ± 1.05 ^#^
EXE + 10SFN	25.97 ± 0.82	26.88 ± 0.82	27.58 ± 0.98 ^#^
EXE + 30SFN	25.65 ± 0.95	26.19 ± 1.58	27.16 ± 1.53
EXE + 90SFN	25.78 ± 1.42	26.98 ± 1.47	27.04 ± 1.43
30SFN	25.66 ± 1.00	26.11 ± 0.97	27.50 ± 1.18 *^,#^

The mice’s ages were 12 weeks old before being gavage fed, 13 weeks old before the EE, and 14 weeks old after the EE. (*: *p* < 0.05 vs. body weight before exercise week; ^#^: *p* < 0.05 vs. body weight before gavage-fed week.)

**Table 2 nutrients-15-03220-t002:** The genes alternated by exercise and returned to normal levels within SFN intervention with the fold change ranked top 10.

Gene ID	Gene Symbol	FDR of Control vs. EXE	Log_2_FC of Control vs. EXE
76487	Ppp1r3g	0.00755186	7.4671905
54698	Crtam	0.00794266	6.65883683
83770	Tas1r2	0.00638426	6.35433732
73435	Tex35	0.01319879	5.87021818
22420	Wnt6	0.03151093	5.42157778
14120	Fbp2	0.00638426	5.36280401
18227	Nr4a2	0.03290719	5.29711301
100502635	Gm11525	0.03715784	5.21183416
15186	Hdc	0.01275262	4.77473087
29870	Gtse1	0.01580759	4.62222956

## Data Availability

The data presented in this study are available upon request from the corresponding author.

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
