# Peer review of "Sulforaphane Inhibits Exhaustive Exercise-Induced Liver Injury and Transcriptome-Based Mechanism Analysis"

_nutrients, 2023, doi:10.3390/nu15143220_

Round 1
Reviewer 1 Report
Jing Yang and colleagues studied sulforaphane inhibits exhaustive exercise-induced liver injury and transcriptome-based mechanism analysis. The result is interesting. The result is interesting, but there are some issues:
1. 3.1. What is the significance of the combined intervention of sulforaphane and EE in slowing incremental weight gain in mice? No age data; SED and EXE are both increased, suggesting SFN is toxic?
2. Figure 2 B, C,Why is the medium dose of SFN the best and the high dose not as good as the medium dose?
3. Table 3 header says ranked top 10, only 6 in the table;
4. Transcriptome experiment was done in the article. But SFN interferes with the EXE mechanism lacks validation experiments.
5. 3.6 and 3.7 title are duplicated.
Reviewer 2 Report
Vigorous exercise may be a health risk, including liver injury, although the underlining mechanisms are not determined. Then the authors investigated the mechanisms of excise-induced hepatic injury in the swimming-excise mice model. The effects of sulforaphane (SFN), a chemical potentially attenuated the exhaustive exercise-induced injury, were examined. They found that exhaustive exercise increased oxidative stress and resulted in hepatic injury. SFN administration reversed these changes. The gene expression analysis suggested that Ppp1r3g, a gene involved in regulating hepatic glycogenesis, plays a critical role in the effects of SFN (and, therefore, in exercise-induced hepatic injury). It is a straightforward study. The results are interesting.
Points to be addressed
1. In lines 85-86, which dose of SFN was used in SED+SFN group? Please label it as SED+90SFN, for instance, if it was 90 mg/kg bw.
2. In Table 3, although the title says ‘ranked top 10’, only 6 genes are listed. Why? Which part of the Venn diagram in Figure 9a corresponds to the genes in Table 3?
3. Have the mRNA expression of the other genes (Crtam, Tas1r2, etc.) been measured by the Q-RT-PCR too? If so, please provide the data in a supplemental Table. It will help readers understand the consistency between RNA-seq and Q-PCR results.
4. Though out the text, Tables, and Figures, group names SED and SED+SFN could be simply put as ‘Control’ and ‘SFN alone.’
